# Incidence and risk factors for heart failure in the ELSA-Brasil cohort

**Ana Paula de Oliveira Lédo**[1¤a☯]*, **Sheila Maria Alvim Matos**[2‡],
**Maria da Conceição Chagas de Almeida**[3‡], **Roque Aras**[1☯]

**1** Graduate Program in Medicine and Health (PPgMS), Faculty of Medicine, Federal University of Bahia (UFBA), Salvador, Bahia, Brazil, **2** Institute of Collective Health (ISC), Federal University of Bahia (UFBA), Salvador, Bahia, Brazil, **3** Gonçalo Muniz Institute, Oswaldo Cruz Foundation (Fiocruz), Salvador, Bahia, Brazil

☯ These authors contributed equally to this work.
‡ These authors also contributed equally to this work.
¤a Current address: Graduate Program in Medicine and Health (PPgMS), Federal University of Bahia (UFBA), Rua Doutor Augusto Viana, S/N – Canela, Salvador, Bahia, Brazil
* anapaulaledo3@gmail.com

## Abstract

### Background

Heart failure (HF) is a clinical condition with high morbidity and mortality and a growing impact on global public health. Longitudinal studies with large samples and extended follow-up are essential to understand its incidence and risk factors in diverse populations.

### Objective

To estimate the incidence over time and identify factors associated with the development of HF among participants of the Longitudinal Study of Adult Health (ELSA-Brasil).

### Methods

A cohort study with 14,854 ELSA-Brasil participants, followed for an average of 12.3 years. A total of 251 individuals with a previous diagnosis of HF at baseline (2008–2010) were excluded. Incident cases were identified in visits 2 (2012–2014) and 3 (2016–2018). Cumulative incidence and incidence rates were estimated. Bivariate comparisons were performed using the chi-square test, while multivariate analysis employed Cox regression to estimate crude and adjusted *Hazard Ratios* (HR) with 95% confidence intervals (*95% CI*). A statistical significance level of *p < 0.05* was adopted.

**Data availability statement:** All relevant data are within the manuscript and its Supporting Information files.

**Funding:** The data used in this study come from the ELSA-Brasil cohort, which is funded by the Brazilian Ministry of Health (Department of Science and Technology) and the Ministry of Science and Technology (FINEP and CNPq), under the following research grant numbers: 01 06 0010.00 RS, 01 06 0212.00 BA, 01 06 0300.00 ES, 01 06 0278.00 MG, 01 06 0115.00 SP, and 01 06 0071.00 RJ. No specific funding was received for the current study. The funders had no role in the design of the study, data collection and analysis, decision to publish, or preparation of the manuscript.

**Competing interests:** The authors have declared that there are no competing interests.

## Results

The incidence of HF was 1.35 per 1,000 person-years (1.39% cumulative) over visits 2 and 3, with 93 new cases in visit 2 (0.61 per 1,000 person-years; 0.63% cumulative) and 113 new cases in visit 3 (0.74 per 1,000 person-years; 0.76% cumulative). Incidence was higher among older adults (65–74 years), self-reported Black individuals, and those with excess weight. Advanced age and abdominal obesity were risk factors present in both visits, while Chagas disease, valvular heart disease, and smoking were specific predictors in visit 2, and hypertension, rheumatic fever, angina, and fatigue were predictors in visit 3.

## Conclusion

The study highlights the progression of heart failure (HF) incidence and identifies important modifiable risk factors in the Brazilian population, reinforcing the need for preventive strategies and public policies focused on early detection and management of HF.

## Introduction

Heart failure (HF) is a clinical syndrome characterized by the heart's inability to adequately meet the body's metabolic demands, often accompanied by increased filling pressures as a compensatory mechanism [1]. It is estimated that 64.3 million people worldwide live with HF [2]. Its prevalence continues to rise due to increased survival, therapeutic advances, and population aging [2, 3, 4]. However, the incidence of HF varies widely across different regions and factors analyzed. According to recent data from the Global Burden of Cardiovascular Diseases and Risks, 1990–2022, published by the World Heart Federation in 2024 [5], the incidence in high-income countries has remained stable or shown a slight decline in recent years. In Europe, incidence rates in 2022 ranged from approximately 2.0 to 3.2 cases per 1,000 person-years, with some regions reporting values below 2.0 per 1,000 person-years [5,6]. In the United States, recent population-based cohorts estimate incidence between 1.7 and 2.2 cases per 1,000 person-years, indicating a decreasing trend compared to previous decades [7,8].

On the other hand, most of the global burden of cardiovascular diseases occurs in low- and middle-income regions, where approximately 80% of cases are concentrated [9]. Factors such as hypertension (HTN), atherosclerosis, obesity, metabolic syndrome, and diabetes mellitus (DM) play a crucial role in increasing the occurrence of HF and acute coronary events [1,8,10]. In these regions, the scarcity and low reliability of data hinder the assessment of the true magnitude of HF. Consequently, the World Health Organization (WHO) has emphasized the urgency of developing more effective strategies for the prevention and control of cardiovascular diseases, particularly in resource-limited countries, where the burden of these conditions is even more significant [11].

In Brazil, HF represents a serious public health problem, associated with high hospitalization rates and significant costs to the healthcare system [12–14]. However, the incidence of the disease remains poorly explored in the country, especially in studies with prospective designs and representative population samples [15]. Available studies are generally limited to specific groups, such as institutionalized elderly, outpatients, or individuals with pre-existing chronic conditions, and often have small samples or cross-sectional designs [16]. These limitations compromise the generalizability of the findings to the adult population at large and hinder the identification of risk factors in the early stages of the disease. This highlights the need for more comprehensive studies, such as ELSA-Brasil, a multicenter prospective cohort study with a large-scale longitudinal design and a long follow-up period, which offers a unique opportunity to generate robust data on the incidence of HF, expand knowledge of its risk factors, and support the development of more effective public health policies. Although the sample is composed of federal civil servants, it presents significant diversity in terms of regions, socioeconomic levels, and races/skin colors within this specific group, allowing the generation of relevant evidence on the occurrence of HF in this occupational context. However, due to the particular characteristics of the sample, generalization of the results to the Brazilian population at large should be approached with caution. The aim of this study was to estimate the incidence of heart failure and identify its predictors among adult participants of the ELSA-Brasil cohort, based on data collected throughout the follow-up period.

## Materials and methods

This prospective cohort study used data from the ELSA-Brasil study, which follows 15,105 federal public servants aged 35–74 years from six educational and research institutions in Brazil: Federal University of Bahia, Federal University of Espírito Santo, Federal University of Minas Gerais, Federal University of Rio Grande do Sul, University of São Paulo, and Oswaldo Cruz Foundation. Recruitment started on August 1, 2008, and ended on January 4, 2021. The primary objective of ELSA-Brasil is to investigate chronic non-communicable diseases and associated risk factors [17,18].

### Study population and follow-up

The present analysis included 14,854 participants from visit 1 (2008–2010), after excluding 251 individuals who reported a prior medical diagnosis of heart failure (HF), as informed in the Previous Medical History (PMH) questionnaire. At baseline, HF identification was based solely on self-report. These participants were followed during visits 2 (2012–2014) and 3 (2016–2018), and extended through 2021. Interviews and in-person examinations were conducted by trained professionals under rigorous quality control [19].

### Data collection

Sociodemographic data, lifestyle habits, and comorbidities were assessed by standardized questionnaires. Blood samples were collected for laboratory tests [20], anthropometric measurements were taken [21], and diagnostic tests, such as electrocardiograms (ECG) [22] and echocardiograms (ECHO) [23], were performed following standardized protocols. Variables were categorized according to physical activity level, alcohol consumption, smoking status, Body Mass Index (BMI), serum triglycerides, and education level. Physical activity levels were classified as follows: vigorous activity (≥150 minutes of vigorous activity per week), moderate activity (≥150 minutes of moderate activity per week), and low activity (10 minutes to <150 minutes of walking or moderate activity per week) [24]. Alcohol consumption was categorized as excessive (>210 g of alcohol per week for men and >140 g per week for women) or non-excessive (<210 g per week for men and <140 g per week for women). BMI categories were defined as obese (≥30 kg/m²), overweight (≥25 and <30 kg/m²), and normal weight (<25 kg/m²) [18]. Education level was divided into two categories: up to high school and higher education.

Information regarding the occurrence and date of death was obtained from the human resources departments of the institutions affiliated with the study, the Mortality Information System (SIM—Ministry of Health), and annual phone calls to monitor participants' health status [25].

## Ethical considerations

This study was conducted in accordance with the principles of the Declaration of Helsinki [26]. The research protocol was approved by the ethics committees of all participating institutions, as well as the National Research Ethics Commission (CEP Registration: 027–06/CEP-ISC). All participants provided written informed consent [27].

## Statistical analysis

Descriptive analyses were performed on the baseline profile of participants from ELSA-Brasil who did not have a prior diagnosis of HF. For continuous variables such as age and waist circumference, the median and interquartile range (Q1: 25th percentile–Q3: 75th percentile) were calculated after verifying normality using the Shapiro-Wilk test. For categorical variables, absolute (n) and relative (%) frequencies were estimated, with comparisons performed using Pearson's chi-square test. The statistical significance level adopted was $p < 0.05$.

Incidence was expressed in two ways: cumulative incidence (or absolute risk), represented by the proportion of new HF cases relative to the initial population, and incidence rate (density), expressed as cases per 1,000 person-years, considering the accumulated follow-up time. To estimate cumulative incidence, only new HF cases identified in visits 2 and 3 at the study's investigation centers were included. The calculation was performed by dividing the number of new HF cases occurring in each period by the total number of individuals free of the condition at the start of follow-up (visit 1). The incidence rate (density) was estimated considering the time until the occurrence of new HF cases, using the Kaplan-Meier method, which allows for the analysis of event-free survival. The life table was applied based on the following criteria: a) Event of interest: new HF cases identified in visits 2 and 3; b) Observation period: 12.3 years (August 1, 2008, to January 4, 2021); c) Censoring: loss to follow-up or participants who did not develop the disease. The comparison between incidence curves was performed using the log-rank test, considering $p < 0.05$ as statistically significant. To identify prognostic factors associated with time to HF development, Cox regression was applied for both univariate and multivariate (adjusted) analyses. Association measures were expressed as Hazard Ratio (HR) and their respective 95% confidence intervals (*95% CI*). Variables included in the multivariate analysis were assessed for the proportional hazards assumption using the test for proportional hazards assumption, adopting $p < 0.05$ as statistically significant.

Missing data were assessed beforehand. As the proportion of missing values was low (<5%) for most variables, analyses were conducted using complete-case analysis (excluding participants with missing information). No imputation methods were applied. All analyses were performed using STATA software, version 16.

## Results

A total of 14,854 baseline participants (2008–2010) from the ELSA-Brasil cohort were evaluated. The sociodemographic and clinical characteristics of this population are presented in **Table 1**. The majority were female, totaling 8,094 (54.5%), with a median age of 51 years (interquartile range: 45–58 years). Regarding lifestyle, 76.8% were classified as minimally active, 40.2% were overweight, and 22.6% were obese. The most prevalent comorbidities were hypertension 5,235 (35.3%) and diabetes mellitus 2,327 (15.7%).

## Incidence

Over 12.3 years of follow-up, 14,854 participants without a prior diagnosis of HF at baseline were monitored. The incidence of HF in the ELSA-Brasil cohort was estimated by identifying new cases in visits 2 and 3. In visit 2 (2012–2014), 93 new cases were identified, considering follow-up losses, observed over a total of 153,133.04 person-years. The cumulative incidence was 0.63%, with an incidence rate of 0.61 per 1,000 person-years (or 61/100,000 person-years). **Table 2** presents the details of the analysis.

In visit 3 (2016–2018), after accounting for follow-up losses, 113 new cases of HF were identified in a total of 153,133.04 person-years of observation. The cumulative incidence was 0.76%, and the incidence rate was 0.74 per 1,000

**Table 1. Sociodemographic and clinical characteristics of participants without a prior diagnosis of HF at visit 1 (n = 14,854).**

| Characteristics | n[†] | Total Participants* (n = 14,854) |
|---|---|---|
| Sex | | |
| Female | 8,094 | (54.5) |
| Male | 6,760 | (45.5) |
| Age (years), median and IQR[‡] | 51 | (45—58) |
| Age range (years) | | |
| 35-44 | 3,315 | (22.3) |
| 45-54 | 5,878 | (39.6) |
| 55-64 | 4,129 | (27.8) |
| 65-74 | 1,519 | (10.2) |
| Race/skin color | | |
| White | 7,695 | (52.4) |
| Brown | 4,127 | (28.1) |
| Black | 2,333 | (15.9) |
| Indigenous | 154 | (1.0) |
| Yellow | 366 | (2.5) |
| Education level | | |
| Higher education (completed) | 7,869 | (53.0) |
| Secondary education (completed) | 6,985 | (47.0) |
| Physical activity | | |
| Highly active | 1,040 | (7.1) |
| Moderately active | 2,356 | (16.0) |
| Low active | 11,245 | (76.8) |
| Smoking status | | |
| Never smoker | 8,483 | (57.1) |
| Former smoker | 4,420 | (29.8) |
| Current smoker | 1,950 | (13.1) |
| Alcohol use | | |
| No | 13,722 | (92.5) |
| Yes | 1,109 | (7.5) |
| BMI (kg/m²) | | |
| Normal weight | 5,522 | (37.1) |
| Overweight | 5,971 | (40.2) |
| Obesity | 3,355 | (22.6) |
| Abdominal obesity | | |
| No | 9,569 | (64.4) |
| Yes | 5,283 | (35.6) |
| Waist circumference (cm), median and IQR [‡] | 90.3 | (82—99) |

ELSA-Brasil (2008–2010).

*Participants without a diagnosis of HF at visit 1. The sum of absolute values may differ due to missing data.

[†]Values presented as absolute frequency (n) and relative frequency (%).

[‡]Values expressed as median and IQR: interquartile range (Q1: 25th percentile – Q3: 75th percentile).

**Table 2. Life table for time in years until the occurrence of HF in visits 2 and 3.**

| Time interval | Total participants visit 2 | New cased visit 2 | Censored observation | Survival probability visit 2 | Total participants visit 3 | New cases visit 3 | Censored obervations | Survival probability visit 3 |
|---|---|---|---|---|---|---|---|---|
| 3–4 years | 13,802 | 1 | 3 | 0.9999 | – | – | – | – |
| 4–5 years | 13,798 | 1 | 53 | 0.9999 | – | – | – | – |
| 5–6 years | 13,744 | 2 | 87 | 0.9997 | – | – | – | – |
| 6–7 years | 13,655 | 2 | 74 | 0.9996 | 12,461 | 0 | 3 | 1.0000 |
| 7–8 years | 13,579 | 1 | 115 | 0.9995 | 12,458 | 0 | 11 | 1.0000 |
| 8–9 years | 13,463 | 6 | 295 | 0.9990 | 12,447 | 3 | 189 | 0.9998 |
| 9–10 years | 13,162 | 17 | 2,977 | 0.9976 | 12,255 | 22 | 2,672 | 0.9977 |
| 10–11 years | 10,168 | 44 | 6,528 | 0.9912 | 9,561 | 54 | 6,108 | 0.9895 |
| 11–12 years | 596 | 19 | 3,518 | 0.9810 | 3,399 | 33 | 3 | 0.9707 |
| 12–13 years | 59 | 0 | 59 | 0.9810 | 56 | 1 | 55 | 0.9367 |

ELSA-Brasil (August 1, 2008, to January 4, 2021).

person-years (or 74/100,000 person-years). According to the life table (**Table 2**), during the 9- to 10-year interval, 12,255 individuals remained at risk, with 22 new cases and 2,672 censored observations. By the end of this interval, 99% of participants had not developed HF.

**Table 3** presents the sociodemographic and clinical characteristics of new HF cases in visits 2 and 3 at the study's investigation centers. In visit 2, HF incidence was similar between men (0.62%) and women (0.63%), being higher among participants aged 65–74 years (1.38%), self-reported Black individuals (1.07%), smokers (0.97%), and former smokers (0.81%). Individuals with overweight (0.30%) and obesity (0.22%) also had a higher incidence, whereas engaging in intense physical activity was associated with a lower occurrence (0.48%).

In visit 3, HF incidence remained similar between sexes, at 0.75% in women and 0.77% in men. However, a considerable increase was observed among participants aged 65–74 years (2.89%), self-reported Black individuals (1.24%), and those with obesity (1.28%). Although intense physical activity continued to show a protective association, excessive alcohol consumption became more strongly related to a higher HF incidence in visit 3 (0.99%).

The crude and adjusted Hazard Ratios (HR) for factors associated with HF incidence in visits 2 and 3 of the ELSA-Brasil cohort, estimated using Cox proportional hazards models, are presented in **Table 4**. In visit 2, in the adjusted model, the factors significantly associated with a higher risk of HF were: abdominal obesity (HR=3.1; *95% CI*: 1.9–4.7), smoking (HR=2.6; *95% CI*: 1.5–4.5), Chagas disease (HR=20.3; *95% CI*: 8.7–47.5), and moderate/severe valvular disease (HR=34.4; *95% CI*: 12.2–96.4). Additionally, for each additional year of age, the risk of HF increased by 6% (HR=1.06; *95% CI*: 1.04–1.09). These findings remained consistent between the crude and adjusted models (*p<0.05*), reinforcing their robustness (**Fig 1**).

In visit 3, in the adjusted model, the factors significantly associated with an increased risk of HF were: overweight (HR=1.9; *95%CI*: 1.1–3.7), abdominal obesity (HR=1.8; *95%CI*: 1.1–3.1), hypertension (HR=1.8; *95%CI*: 1.2–2.7), fatigue symptoms (HR=1.6; *95%CI*: 1.1–2.3), angina (HR=2.6; *95%CI*: 1.4–4.9), and rheumatic fever (HR=2.6; *95%CI*: 1.3–5.2). The risk of HF also increased progressively with age, with a 3% increase per additional year (HR=1.03; *95%CI*: 1.01–1.06). Self-reported Black race/skin color was significantly associated in the univariate analysis (HR=1.7; *95%CI*: 1.1–2.7) but lost significance after adjustment for confounding factors (HR=1.4; *95%CI*: 0.9–2.3).

The total HF incidence estimate was obtained through the combined analysis of visits 2 and 3, reflecting disease progression in the cohort over 12.3 years. Among the 14,854 baseline participants, 206 new HF cases were identified after accounting for losses to follow-up, resulting in a total observation time of 306,266.08 person-years. The cumulative incidence was 1.39%, and the incidence rate was 1.35 per 1,000 person-years (or 135/100,000 person-years) (**Table 5**).

**Table 3. Population distribution and heart failure incidence estimates in Visits 2 and 3, according to sociodemographic and clinical characteristics of participants without heart failure at baseline.**

| Characteristics* | n† (%) | Person-years at Risk visit 2 | New cases visit 2 | Cumulative Incidence (%) visit 2 | HF Incidence rate per 1,000 person-years visit 2 | Person-years at Risk visit 3 | New cases visit 3 | Cumulative Incidence (%) visit 3 | HF Incidence rate per 1,000 person-years visit 3 |
|---|---|---|---|---|---|---|---|---|---|
| Sex | | | | | | | | | |
| Female | 8,094 (54.5) | 51,133.04 | 51 | (0.63) | 0.60 | 32,376 | 61 | (0.75) | 1.88 |
| Male | 6,760 (45.5) | 68,820.40 | 42 | (0.62) | 0.61 | 27,040 | 52 | (0.77) | 1.92 |
| Age range (years) | | | | | | | | | |
| 35–44 | 3,315 (22.3) | 13,461.7 | 7 | (0.21) | 0.52 | 13,260 | 1 | (0.03) | 0.07 |
| 45–54 | 5,878 (39.6) | 33,559.64 | 33 | (0.56) | 0.98 | 23,512 | 22 | (0.37) | 0.93 |
| 55–64 | 4,129 (27.8) | 39,346.27 | 32 | (0.77) | 0.81 | 16,516 | 38 | (0.92) | 2.30 |
| 65–74 | 1,519 (10.2) | 58,285.03 | 21 | (1.38) | 0.36 | 6,076 | 44 | (2.89) | 6.75 |
| Race/skin color | | | | | | | | | |
| White | 7,695 (52.4) | 62,856.15 | 37 | (0.48) | 0.59 | 30,780 | 55 | (0.72) | 1.79 |
| Brown | 4,127 (28.1) | 33,69.62 | 29 | (0.70) | 0.86 | 16,508 | 26 | (0.63) | 1.57 |
| Black | 2,333 (15.9) | 19,033.68 | 25 | (1.07) | 1.31 | 9,332 | 29 | (1.24) | 3.11 |
| BMI (kg/m²) | | | | | | | | | |
| Normal weight | 5,522 (37.1) | 44,502.73 | 16 | (0.11) | 0.36 | 22,088 | 17 | (0.31) | 0.77 |
| Overweight | 5,971 (40.2) | 48,221.28 | 44 | (0.30) | 0.91 | 23,884 | 53 | (0.89) | 2.22 |
| Obesity | 3,355 (22.6) | 27,109.48 | 33 | (0.22) | 1.22 | 13,420 | 43 | (1.28) | 3.20 |
| Physical activity | | | | | | | | | |
| High | 1,040 (7.1) | 8,839.06 | 5 | (0.48) | 0.57 | 4,160 | 7 | (0.67) | 1.68 |
| Moderate | 2,356 (16.0) | 19,021.91 | 13 | (0.55) | 0.68 | 9,424 | 16 | (0.68) | 1.70 |
| Low | 11,245 (76.8) | 92,092.49 | 74 | (0.66) | 0.80 | 44,980 | 88 | (0.78) | 1.96 |
| Smoking Status | | | | | | | | | |
| Never Smoker | 8,483 (57.1) | 51,177.94 | 38 | (0.45) | 0.74 | 33,932 | 67 | (0.79) | 1.97 |
| Former Smoker | 4,420 (29.8) | 26,665.86 | 36 | (0.81) | 1.35 | 17,680 | 31 | (0.70) | 1.75 |
| Current Smoker | 1,950 (13.1) | 11,764.35 | 19 | (0.97) | 1.62 | 7,800 | 15 | (0.77) | 1.92 |
| Alcohol use | | | | | | | | | |
| No | 13,722 (92.5) | 93,381.74 | 86 | (0.63) | 0.92 | 54,888 | 102 | (0.74) | 1.86 |
| Yes | 1,109 (7.5) | 6,973.59 | 7 | (0.64) | 1.00 | 4,436 | 11 | (0.99) | 2.48 |

ELSA-Brasil (2008/2010–2012/2014; 2008/2010–2016/2018).

*Characteristics of participants without a heart failure diagnosis at baseline. The sum of absolute values may vary due to missing data.

†Values are presented as absolute frequency (n) and relative frequency.

## Discussion

We observed a progressive increase in the incidence of HF over 12.3 years of follow-up in the ELSA-Brasil cohort. Incidence rates per 1,000 person-years were 0.61 at visit 2 and 0.74 at visit 3, resulting in a total cumulative incidence of 1.39% and an average rate of 1.35 per 1,000 person-years (135/100,000 person-years). These findings confirm the epidemiological relevance of the cohort, especially given the scarcity of longitudinal HF studies in Brazil. Moreover, they are consistent with South American studies reporting rates close to 1.99 per 1,000 person-years [15,16], supporting the robustness and applicability of our results in similar contexts.

**Table 4. Cox proportional hazards regression for the incidence of HF in visits 2 (n = 93) and 3 (n = 113) of the ELSA-Brasil cohort.**

| Variables | Crude HR*<br>95% CI**<br>(visit 2) | Adjusted HR<br>95% CI<br>(visit 2) | Crude HR<br>95% CI<br>(visit 3) | Adjusted HR<br>95% CI<br>(visit 3) |
|---|---|---|---|---|
| Age (years) | 1.07 (1.05—1.10) | 1.06 (1.04—1.09) | 1.05 (1.03—1.07) | 1.03(1.01—1.06) |
| Abdominal obesity | | | | |
| No | 1 | | 1 | |
| Yes | 3.13 (2.06—4.76) | 3.06 (1.99—4.71) | 3.32 (2.26—4.87) | 1.83 (1.10—3.06) |
| Smoking status | | | | |
| Never smoker | 1 | | – | – |
| Former smoker | 1.85 (1.17—2.92) | 1.5(0.98—2.50) | | |
| Current smoker | 2.22 (1.28—3.86) | 2.59(1.48—4.52) | | |
| Left ventricular dilation (>6.5 cm) | | | | |
| No | 1 | | | |
| Yes | 192.64(47.14—787.22) | 4.63(0.78—27.67) | | |
| Chagas disease | | | | |
| No | 1 | | | |
| Yes | 20.98 (9.16—48.04) | 20.32(8.70—47.46) | | |
| Valvulopathy (moderate/severe) | | | | |
| No | 1 | | | |
| Yes | 49.54(21.60—113.61) | 34.37(12.25—96.38) | | |
| Race/skin color | | | | |
| White | – | – | 1 | |
| Brown | – | – | 1.73 (1.10—2.72) | 1.43 (0.90—2.26) |
| Black | – | – | 0.84 (0.53—1.35) | 0.82 (0.52—1.32) |
| Others‡ | – | – | 0.26 (0.36—1.87) | 0.29 (0.40—2.10) |
| BMI (kg/m²) | | | | |
| Normal weight | | | 1 | |
| Overweight | | | 2.98 (1.72—5.14) | 1.99 (1.08—3.69) |
| Obesity | | | 4.31 (2.45—7.56) | 1.88 (0.89—3.96) |
| Hypertension | | | | |
| No | – | – | 1 | |
| Yes | – | – | 2.85 (1.96—4.15) | 1.81 (1.20—2.74) |
| Fatigue symptom | | | | |
| No | – | – | 1 | |
| Yes | – | – | 1.74 (1.21—2.53) | 1.58 (1.08—2.32) |
| Angina | | | | |
| No | – | – | 1 | |
| Yes | – | – | 4.71(2.59—8.58) | 2.62 (1.41—4.86) |
| Rheumatic fever | | | | |
| No | – | – | 1 | |
| Yes | – | – | 2.91 (1.47—5.75) | 2.61(1.31—5.19) |

ELSA-Brasil (August 1, 2008, to January 4, 2021);

*HR: *hazard ratios*.

**95%CI*: 95% confidence interval.

‡Others: (Indigenous and Yellow).

†p: corresponding to the log-rank test, with statistical significance: *p-value <0.05*.

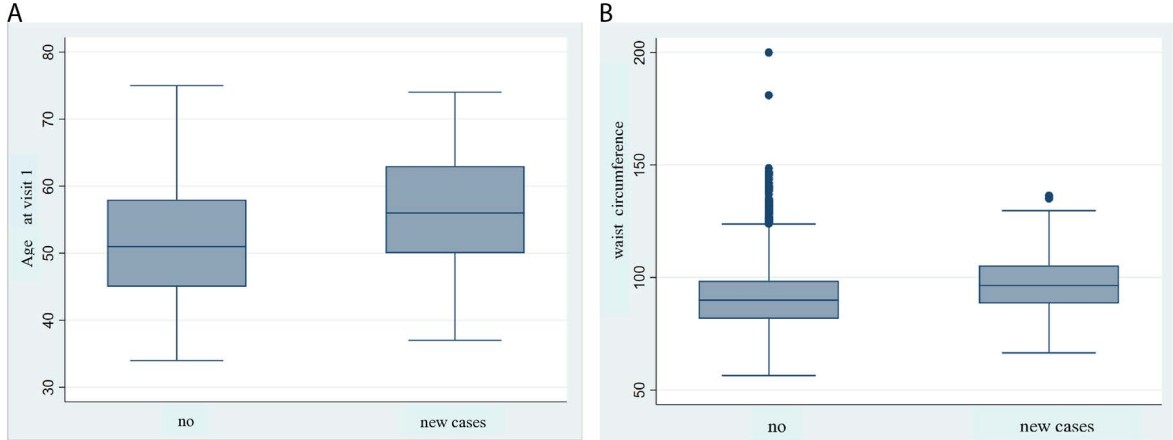

**Fig 1. Association of baseline age and waist circumference with incident heart failure in ELSA-Brasil.** (A) Box plot showing baseline age comparing participants without heart failure (HF) (n = 14,854) and incident HF cases at visit 2 (n = 93). Median age was significantly higher among incident cases ($p < 0.01$), period 2008–2014. (B) Box plot showing baseline waist circumference comparing participants without HF (n = 14,854) and incident HF cases at visit 3 (n = 113). Waist circumference was significantly greater among incident cases ($p < 0.01$), period 2008–2018.

**Table 5. Total Incidence Results of HF in Visits 2 and 3.**

| Period | New cases of HF | Cumulative Incidence (%) | Incidence rate per 1,000 person-years |
|---|---|---|---|
| Visit 2 (2012–2014) | 93 | (0.63) | 0.61 |
| Visit 3 (2016–2018) | 113 | (0.76) | 0.74 |
| Total | 206 | (1.39) | 1.35 |

ELSA-Brasil (August 1, 2008, to January 4, 2021)

HF incidence varies by region and population profile [5–7]. In high-income countries, there is a trend toward stabilization or decline in incidence rates [2,3], whereas data from Latin America remain scarce and heterogeneous. For example, cohort studies conducted in Argentina have reported rates ranging from 1.37 to 5.57 per 1,000 person-years, highlighting regional disparities and underscoring the need for health policies based on local data [16].

In the ELSA-Brasil cohort, the baseline population was predominantly female (54.5%) with a median age of 51 years. There was a high prevalence of overweight, physical inactivity, hypertension, and diabetes, conditions widely recognized as risk factors for HF due to their contribution to systemic inflammation, ventricular dysfunction, and cardiac remodeling [1,2,8,10].

No statistically significant difference in HF incidence was observed between men and women during follow-up ($p = 0.44$). Although previous studies report a higher risk among men, particularly in cases of heart failure with reduced ejection fraction (HFrEF) [28], recent evidence suggests that shared clinical, social, and behavioral factors may mitigate this difference in populations with equitable access to healthcare, such as ours [1,2,10]. This finding highlights the importance of considering both the different clinical presentations of HF and the social determinants of health when interpreting data.

Cox regression models identified advanced age and abdominal obesity as consistent and significant predictors of HF risk at visits 2 and 3, underscoring their central role in the disease's pathophysiology. Aging induces progressive structural and functional cardiac changes, especially when accompanied by comorbidities and degenerative senescence processes [4,8]. Abdominal obesity contributes to chronic inflammation, insulin resistance, and endothelial dysfunction—key mechanisms in HF development [2,9].

At visit 2, smoking, Chagas disease, and valvular heart disease were significantly associated with HF incidence. Smoking is a well-established cardiovascular risk factor due to its inflammatory, pro-thrombotic, and oxidative effects [10]. Chagas disease and valvular pathologies, both endemic conditions in Brazil, represent relevant structural causes of HF with significant clinical and prognostic impact [4,29,30].

At visit 3, in addition to previously identified predictors, new clinical factors and symptoms emerged, including hypertension, overweight, fatigue, angina, and rheumatic fever. Hypertension is a central determinant of the disease due to its role in inducing hypertrophy and ventricular dysfunction [10]. Overweight contributes to metabolic dysfunction, increased hemodynamic load, and systemic inflammation [2,9]. Symptoms such as fatigue and angina may indicate early clinical manifestations, emphasizing the importance of early screening and structured follow-up [29–32]. Although less prevalent, rheumatic fever evidences the persistence of infectious causes in the Brazilian context.

In univariate analysis at visit 3, self-reported Black individuals exhibited a higher risk of HF, but this association lost significance after adjustment for socioeconomic and clinical factors, suggesting disparities reflect structural inequalities and the impact of structural racism rather than skin color per se [1,5,8]. This underscores the urgent need to incorporate racial equity into preventive strategies and strengthen policies addressing social inequalities disproportionately affecting the Black population.

The findings from the ELSA-Brasil cohort reveal the dynamic evolution of factors associated with HF over time and highlight the need for integrated prevention strategies focused on rigorous control of modifiable risk factors and early screening. In Brazil, public policies such as income transfer programs and initiatives aimed at strengthening primary care for hypertension and diabetes management have shown potential to modify the population's epidemiological profile and should be fully integrated into HF control strategies [33,34].

## Limitations

The cohort is composed of federal public servants, which may limit representativeness of the socioeconomic extremes and attenuate associations with certain risk factors. There is potential for healthy worker bias, reliance on self-reported variables, and loss to follow-up. Despite the long average follow-up (12.3 years), the number of incident cases aligns with the chronic, slowly progressive nature of HF and may not fully capture late-onset case

## Conclusions

The results of this study highlight the incidence of heart failure (HF) in the ELSA-Brasil cohort, with higher occurrence among older adults, individuals with obesity, and smokers. A progressive increase in incidence rates was observed over time. Advanced age and abdominal obesity were associated with HF in both follow-up phases, while Chagas disease, valvular heart disease, and smoking were significant predictors in Visit 2, and hypertension in Visit 3. These findings underscore the importance of early detection and appropriate management of cardiovascular conditions, as well as the control of modifiable risk factors. Such evidence can inform public policies and more effective prevention strategies to reduce the burden of HF in Brazil.

## Supporting information

**S1 File. Analysis of Age and Waist Circumference in Incident Heart Failure Cases.** This file contains box plots and statistical analyses of baseline age and waist circumference comparing participants with and without incident heart failure at visits 2 and 3 of the ELSA-Brasil cohort. It includes tests of normality (Shapiro-Wilk), justification for using non-parametric tests (Mann-Whitney), and presentation of median values.
(DOCX)

## Acknowledgments

We thank the cohort participants who agreed to collaborate in this study and the ELSA-Brasil research team for their valuable contributions.

This article is part of the doctoral thesis of Ana Paula de Oliveira Lédo, developed within the Graduate Program in Medicine and Health, School of Medicine, Federal University of Bahia, Salvador, Bahia, Brazil.

## Author contributions

**Conceptualization:** Ana Paula de Oliveira Lédo, Roque Aras.

**Data curation:** Ana Paula de Oliveira Lédo.

**Formal analysis:** Ana Paula de Oliveira Lédo.

**Investigation:** Ana Paula de Oliveira Lédo, Sheila Maria Alvim Matos, Maria da Conceição Chagas Almeida.

**Methodology:** Ana Paula de Oliveira Lédo, Sheila Maria Alvim Matos, Maria da Conceição Chagas Almeida, Roque Aras.

**Project administration:** Roque Aras.

**Supervision:** Sheila Maria Alvim Matos, Maria da Conceição Chagas Almeida, Roque Aras.

**Writing – original draft:** Ana Paula de Oliveira Lédo.

**Writing – review & editing:** Ana Paula de Oliveira Lédo, Sheila Maria Alvim Matos, Maria da Conceição Chagas Almeida, Roque Aras.

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
