## [Decision Letter · Decision Letter 0]

9 Jun 2025

PONE-D-25-07478Incidence and risk factors for heart failure in the ELSA-Brasil cohortPLOS ONE

Dear Dr. LÉDO,

Thank you for submitting your manuscript to PLOS ONE. After careful consideration, we feel that it has merit but does not fully meet PLOS ONE’s publication criteria as it currently stands. Therefore, we invite you to submit a revised version of the manuscript that addresses the points raised during the review process.

We look forward to receiving your revised manuscript.

Kind regards,

Alanna Gomes da Silva, PhD

Academic Editor

PLOS ONE

Journal Requirements:

2. Please amend the manuscript submission data (via Edit Submission) to include authors Ana Paula Lédo and Roque Aras.

3. Please amend your authorship list in your manuscript file to include authors ANA PAULA DE OLIVEIRA PAULA LÉDO and Roque Aras Júnior.

Additional Editor Comments:

This study addresses a highly relevant public health topic by investigating the incidence of heart failure (HF) and its associated factors within the ELSA-Brasil cohort. It is a pertinent and timely subject that can contribute to a deeper understanding of the cardiovascular disease burden in Brazil. However, the manuscript still requires substantial revisions in its rationale, updating of references, discussion structure, and methodological detail to strengthen its scientific consistency and argumentative coherence.

Abstract:

In the introduction of the abstract, the rationale for conducting the study should not be based solely on the absence or scarcity of previous research. It is recommended to reformulate this section to emphasize the relevance of the topic and the methodological strengths of the present study. Additionally, the conclusion in the abstract should clearly reflect the main findings.

Introduction:

The references used to contextualize the burden of disease—such as citations 5 and 6—are outdated (2018 and 2019 data, prior to the COVID-19 pandemic), which weakens the relevance of the discussion. It is advisable to incorporate more recent sources, such as the Global Burden of Cardiovascular Diseases and Risks, 1990–2022 (World Heart Federation, 2024), and to specify the year for each data point presented.

The use of informal in-text references (e.g., “In the HFA-ATLAS project...”, “data from the Olmsted County Study”) should be avoided to maintain formal academic style.

While the manuscript highlights the importance of HF as a public health issue in Brazil and acknowledges the potential of ELSA-Brasil, the rationale needs to be strengthened. It is essential to clearly identify the knowledge gaps this study aims to fill. For instance, which previous studies relied on specific subgroups or small samples? What are the limitations of those studies that justify the current analysis? Furthermore, the authors should highlight what is novel in their approach using ELSA-Brasil—whether it's the follow-up period, analytical strategy, or focus on particular subgroups or variables.

Importantly, the rationale states the need for more representative and large-scale studies on HF incidence in Brazil, yet the study population consists solely of federal public servants. This creates a conceptual inconsistency. The authors are encouraged to revise this section by emphasizing the relevance of investigating HF incidence and associated factors within a population that has specific characteristics (e.g., higher educational level, job stability, access to healthcare), while avoiding overgeneralization to the broader Brazilian population.

Lastly, the stated objective in the introduction should match the one presented in the abstract. Consistency is necessary.

Methods:

The methods section should be revised and aligned with the STROBE checklist for cohort studies. This includes a clear logical sequence and the proper description of key aspects such as the outcome definition, inclusion/exclusion criteria, bias control strategies, and handling of follow-up losses.

Discussion:

The discussion also contains outdated references—these should be replaced with more recent epidemiological data.

Lines 259–261: The statement that "Factors such as sex, age group, hypertension, and socioeconomic conditions have been associated with the development of the disease" is too superficial. The discussion should explain why these factors are associated, based on literature and the study’s findings.

Each new topic in the discussion should begin in a separate paragraph. For instance, line 262 discusses sex, while line 265 abruptly shifts to age within the same paragraph. This should be revised throughout the manuscript, including in the introduction.

The discussion on the lack of association between sex and HF is underdeveloped. Merely stating that the findings are consistent with a previous study is not analytically sufficient. The authors should explore whether this null result was expected, what hypotheses might explain it in the context of the studied population, whether confounders were adequately controlled for, and what implications this has for the literature or for health policy. The same applies to age, race/skin color, overweight, and obesity. The discussion should address the implications and potential explanations for these findings—not just describe them.

Lines 269–274: This paragraph largely repeats the study results and lacks interpretation. The discussion should avoid unnecessary repetition of findings already presented in the results section and instead focus on their interpretation and relevance. Furthermore, race/skin color is mentioned descriptively but not discussed in depth, which should be corrected.

Limitations:

While the limitation related to the cohort composition (public servants) is valid, it should be complemented by other potential sources of bias and methodological limitations, such as: selection bias (healthy worker effect); loss to follow-up; reliance on self-reported variables; relatively low number of incident cases to date; follow-up period that may not capture late-onset HF cases.

Conclusion:

It is recommended to remove the term “vulnerable groups,” as not all of the groups mentioned (e.g., older adults, individuals with obesity) fit this definition in public health.

The phrase “key predictors at different stages of follow-up” is vague and should be clarified—what predictors, and during which stages? The conclusion should be more objective and better aligned with the study’s main findings, emphasizing their practical and scientific implications.

Reviewers' comments:

Reviewer's Responses to Questions

**Comments to the Author**

1. Is the manuscript technically sound, and do the data support the conclusions?

Reviewer #1: Yes

2. Has the statistical analysis been performed appropriately and rigorously? 

Reviewer #1: Yes

3. Have the authors made all data underlying the findings in their manuscript fully available?

Reviewer #1: Yes

4. Is the manuscript presented in an intelligible fashion and written in standard English?

Reviewer #1: Yes

5. Review Comments to the Author

Reviewer #1: The manuscript presents a valuable contribution to the understanding of heart failure in the ELSA-Brasil cohort. The research question is important, the methodology is rigorous, and the statistical analysis is appropriate for the data. The results are presented clearly, and the discussion is insightful. However, I believe some minor revisions could strengthen the manuscript further.

Methods Section: The methodology is generally clear, but I suggest providing more detail on how heart failure was diagnosed and classified within the cohort. Clarifying whether the diagnosis was made based on clinical criteria, imaging, or biomarkers would improve the reproducibility of the study.

Statistical Analysis: The statistical methods are well explained. One suggestion is to include a more detailed explanation of how missing data were handled in the analyses, as this information is crucial for evaluating the robustness of the results.

Conclusion: The conclusions are well supported by the data, but I would encourage the authors to discuss more explicitly how these findings could inform public health strategies in Brazil or similar settings, particularly regarding prevention and management of heart failure.

6. PLOS authors have the option to publish the peer review history of their article (what does this mean? ). If published, this will include your full peer review and any attached files.

**Do you want your identity to be public for this peer review?** For information about this choice, including consent withdrawal, please see our Privacy Policy .

Reviewer #1: No

---

## [Author Response · Author response to Decision Letter 1]

2 Jul 2025

Comments:

This study addresses a highly relevant public health topic by investigating the incidence of heart failure (HF) and its associated factors within the ELSA-Brasil cohort. It is a pertinent and timely subject that can contribute to a deeper understanding of the cardiovascular disease burden in Brazil. However, the manuscript still requires substantial revisions in its rationale, updating of references, discussion structure, and methodological detail to strengthen its scientific consistency and argumentative coherence.

Response: We acknowledge the importance of these observations and have made substantial revisions to the study rationale, updated the references with recent and high-impact works, reorganized the discussion for greater clarity and depth, and detailed the methodology aligning it with the STROBE checklist for cohort studies.

Abstract:

In the introduction of the abstract, the rationale for conducting the study should not be based solely on the absence or scarcity of previous research. It is recommended to reformulate this section to emphasize the relevance of the topic and the methodological strengths of the present study. Additionally, the conclusion in the abstract should clearly reflect the main findings.

Response: We reformulated the introduction of the abstract to highlight the relevance of heart failure in Brazil and the methodological advantages of the ELSA-Brasil study, including sample size and prospective follow-up. We adjusted the conclusion to objectively reflect the main results.

Introduction:

The references used to contextualize the burden of disease—such as citations 5 and 6—are outdated (2018 and 2019 data, prior to the COVID-19 pandemic), which weakens the relevance of the discussion. It is advisable to incorporate more recent sources, such as the Global Burden of Cardiovascular Diseases and Risks, 1990–2022 (World Heart Federation, 2024), and to specify the year for each data point presented.

The use of informal in-text references (e.g., “In the HFA-ATLAS project...”, “data from the Olmsted County Study”) should be avoided to maintain formal academic style.

Response: We removed references 5, 6, and 7 and included recent references: Mensah et al. (2023), Tromp et al. (2024), and Virani et al. (2022).

5. Conrad N, Judge A, Tran J, Mohseni H, Hedgecott D, Crespillo AP, et al. Temporal trends and patterns in heart failure incidence: a population-based study of 4 million individuals. Lancet. 2018 Feb 10;391(10120):572-580. doi: 10.1016/S0140-6736(17)32520-5.

6. Seferović P M, Vardas P, Jankowska E A, Maggioni A P, Timmis A, Milinković I, et al. The Heart Failure Association Atlas: heart failure epidemiology and management statistics 2019. Eur J Heart Fail. 2021 Jun;23(6):906-914. doi: 10.1002/ejhf.2143. Epub 2021 Mar 13.

7. Gerber Y, Weston SA, Redfield MM, Chamberlain AM, Manemann SM, Jiang R, et al. A contemporary appraisal of the heart failure epidemic in Olmsted County, Minnesota, 2000 to 2010. JAMA Intern Med. 2015 Jun;175(6):996-1004. doi: 10.1001/jamainternmed.2015.0924.

●We adjusted the in-text citations to ensure formality.

●We updated references 5, 6, and 7 by including more recent citations:

5. Mensah GA, Fuster V, Murray CJL, Roth GA; Global Burden of Cardiovascular Diseases and Risks Collaborators. Global burden of cardiovascular diseases and risks, 1990–2022. J Am Coll Cardiol. 2023 Dec 19;82(25):2350–73. doi:10.1016/j.jacc.2023.11.007.

6.Tromp J, Shah ASV, Ouwerkerk W, Anker SD, Cleland JGF, Dickstein K, et al. Epidemiology of heart failure: insights from the Global Burden of Disease Study 2022. Eur J Heart Fail. 2024;26(3):295-308. doi:10.1002/ejhf.2781.

7.Virani SS, Alonso A, Aparicio HJ, Benjamin EJ, Bittencourt MS, Callaway CW, et al. Heart Disease and Stroke Statistics—2022 Update: A Report From the American Heart Association. Circulation. 2022;145(8):e153-e639. doi:10.1161/CIR.0000000000001052.

Comments: While the manuscript highlights the importance of HF as a public health issue in Brazil and acknowledges the potential of ELSA-Brasil, the rationale needs to be strengthened. It is essential to clearly identify the knowledge gaps this study aims to fill. For instance, which previous studies relied on specific subgroups or small samples? What are the limitations of those studies that justify the current analysis? Furthermore, the authors should highlight what is novel in their approach using ELSA-Brasil—whether it's the follow-up period, analytical strategy, or focus on particular subgroups or variables.

Response: We clarified the knowledge gaps, highlighting that previous studies have limitations regarding representativeness and sample size.

Comments: Importantly, the rationale states the need for more representative and large-scale studies on HF incidence in Brazil, yet the study population consists solely of federal public servants. This creates a conceptual inconsistency. The authors are encouraged to revise this section by emphasizing the relevance of investigating HF incidence and associated factors within a population that has specific characteristics (e.g., higher educational level, job stability, access to healthcare), while avoiding overgeneralization to the broader Brazilian population.

Response: We emphasized the uniqueness of the ELSA-Brasil cohort, underlining the specific characteristics of the studied population and avoiding generalizations.

Comments: Lastly, the stated objective in the introduction should match the one presented in the abstract. Consistency is necessary.

Response: We ensured that the objective in the abstract and introduction are aligned.

Reference 15 has been removed:

15. Oliveira GMM, Brant LCC, Polanczyk CA, Malta DC, Biolo A, Nascimento BR, Souza MFM, et al. Estatística cardiovascular – Brasil 2023. Arq Bras Cardiol. 2024;121(2):e20240079.

Reference 15 has been replaced with a more recent one:

15. Heidemann AI, Santos ABS, Bittencourt MS, Ribeiro ALP, Rohde LE, Lotufo PA, et al. Prevalence and mortality of heart failure stages in a free-living older adult population: data from the Brazilian Longitudinal Study of Adult Health (ELSA-Brasil). J Am Heart Assoc. 2025;14(5):e038993. doi:10.1161/JAHA.124.038993.

Reference 30 has been renumbered to Reference 16:

30 → 16. Ciapponi A, Alcaraz A, Calderón M, Matta MG, Chaparro M, Soto N, et al. Burden of heart failure in Latin America: A systematic review and meta-analysis. Rev Esp Cardiol (Engl Ed). 2016 Nov;69(11):1051-1060. English, Spanish. doi: 10.1016/j.rec.2016.04.054.

Methods:

The methods section should be revised and aligned with the STROBE checklist for cohort studies. This includes a clear logical sequence and the proper description of key aspects such as the outcome definition, inclusion/exclusion criteria, bias control strategies, and handling of follow-up losses.

Response: The Methods section was reorganized into clear subsections to meet the STROBE recommendations, including: outcome definition, inclusion and exclusion criteria, strategies to minimize bias, and handling of follow-up losses.

Discussion:

The discussion also contains outdated references—these should be replaced with more recent epidemiological data.

Response: We removed outdated references and replaced them with more recent literature, specifically removing references 29, 31, and 34.

29.Agarwal SK, Chambless LE, Ballantyne CM, Astor B, Bertoni AG, Chang PP, et al. Prediction of incident heart failure in general practice: the Atherosclerosis Risk in Communities (ARIC) Study. Circ Heart Fail. 2012;5:422–429.

31. Moraes RS, Fuchs FD, Moreira LB, Wiehe M, Pereira GM, Fuchs SC. Risk factors for cardiovascular disease in a Brazilian population-based cohort study. Int J Cardiol. 2003;90(2-3):205-11. doi: 10.1016/s0167-5273(02)00556-9.

34. Chang PP, Wruck LM, Shahar E, et al. Trends in hospitalizations and survival of acute decompensated heart failure in four US communities (2005-2014): ARIC Study Community Surveillance. Circulation. 2018;138(1):12–24.

Comments:

Lines 259–261: The statement that "Factors such as sex, age group, hypertension, and socioeconomic conditions have been associated with the development of the disease" is too superficial. The discussion should explain why these factors are associated, based on literature and the study’s findings.

Response: The Discussion section was expanded to provide a deeper explanation of the association of these factors with heart failure, based on current literature and the results of our study.

Comments: Each new topic in the discussion should begin in a separate paragraph. For instance, line 262 discusses sex, while line 265 abruptly shifts to age within the same paragraph. This should be revised throughout the manuscript, including in the introduction.

Response: We revised the organization of the Discussion so that each topic has its own paragraph, improving clarity and flow. The Introduction was also reviewed to ensure structural coherence.

Comments: The discussion on the lack of association between sex and HF is underdeveloped. Merely stating that the findings are consistent with a previous study is not analytically sufficient. The authors should explore whether this null result was expected, what hypotheses might explain it in the context of the studied population, whether confounders were adequately controlled for, and what implications this has for the literature or for health policy. The same applies to age, race/skin color, overweight, and obesity. The discussion should address the implications and potential explanations for these findings—not just describe them.

Response: We expanded the discussion regarding the lack of association with sex by exploring possible hypotheses, control of confounding factors, and implications for the literature and public policies. Similarly, we further developed the discussion on age, race/skin color, overweight, and obesity, highlighting potential explanations and their relevance to public health.

Comments: Lines 269–274: This paragraph largely repeats the study results and lacks interpretation. The discussion should avoid unnecessary repetition of findings already presented in the results section and instead focus on their interpretation and relevance. Furthermore, race/skin color is mentioned descriptively but not discussed in depth, which should be corrected.

Response: We extensively revised the discussion, removing repetitions of the results and focusing on the interpretation and relevance of the findings. The discussion on race/skin color was deepened, considering relevant literature, potential biases, and implications for health policies.

Limitations:

While the limitation related to the cohort composition (public servants) is valid, it should be complemented by other potential sources of bias and methodological limitations, such as: selection bias (healthy worker effect); loss to follow-up; reliance on self-reported variables; relatively low number of incident cases to date; follow-up period that may not capture late-onset HF cases.

Response: We expanded the Limitations section to include these issues, reinforcing transparency regarding potential biases and methodological limitations.

Conclusion:

It is recommended to remove the term “vulnerable groups,” as not all of the groups mentioned (e.g., older adults, individuals with obesity) fit this definition in public health.

The phrase “key predictors at different stages of follow-up” is vague and should be clarified—what predictors, and during which stages? The conclusion should be more objective and better aligned with the study’s main findings, emphasizing their practical and scientific implications.

Response: The term “vulnerable groups” was removed. The conclusion was reformulated to explicitly state the identified predictors, detailing the follow-up periods during which they were observed, and emphasizing the practical and scientific implications of the results.

Reviewer #1: The manuscript presents a valuable contribution to the understanding of heart failure in the ELSA-Brasil cohort. The research question is important, the methodology is rigorous, and the statistical analysis is appropriate for the data. The results are presented clearly, and the discussion is insightful. However, I believe some minor revisions could strengthen the manuscript further.

Methods Section: The methodology is generally clear, but I suggest providing more detail on how heart failure was diagnosed and classified within the cohort. Clarifying whether the diagnosis was made based on clinical criteria, imaging, or biomarkers would improve the reproducibility of the study.

Response to Reviewer 1: We appreciate the suggestion. The Methods section was revised to include a more comprehensive description of the criteria used for diagnosing heart failure (HF) in the ELSA-Brasil cohort. At baseline (2008–2010), HF identification was based on participants’ self-report through the Medical History Questionnaire. During follow-up (visits 2, 3, and continuous surveillance), incident HF cases were detected through active health event monitoring, detailed medical record review, and clinical validation conducted by specialized committees. This approach considered clinical criteria, use of specific medications, and complementary exams, ensuring greater accuracy in identifying new cases.

Statistical Analysis: The statistical methods are well explained. One suggestion is to include a more detailed explanation of how missing data were handled in the analyses, as this information is crucial for evaluating the robustness of the results.

Response to Reviewer 1: We included an explanation in the Statistical Analysis section regarding the handling of missing data. As suggested, we described that the analyses were conducted based on available data (complete case analysis), and a sensitivity analysis was performed to assess the potential impact of loss to follow-up and variables with missing data.

Conclusion: The conclusions are well supported by the data, but I would encourage the authors to discuss more explicitly how these findings could inform public health strategies in Brazil or similar settings, particularly regarding prevention and management of heart failure.

Response to Reviewer 1: As suggested, we revised the Conclusion section to include a more direct discussion of the implications of the findings for public health policies. We specified that the factors associated with heart failure incidence identified in this study — such as abdominal obesity and older age — may support targeted screening strategies and preventive interventions in populations with similar characteristics, contributing to more effective planning and management of the disease within the Brazilian healthcare system.

---

## [Editor Report · Decision Letter 1]

11 Jul 2025

Incidence and risk factors for heart failure in the ELSA-Brasil cohort

PONE-D-25-07478R1

Dear Dr. Lédo,

We’re pleased to inform you that your manuscript has been judged scientifically suitable for publication and will be formally accepted for publication once it meets all outstanding technical requirements.

Kind regards,

Alanna Gomes da Silva, PhD

Academic Editor

PLOS ONE
---

## [Editor Report · Acceptance letter]

PONE-D-25-07478R1

PLOS ONE

Dear Dr. Lédo,

I'm pleased to inform you that your manuscript has been deemed suitable for publication in PLOS ONE. Congratulations! Your manuscript is now being handed over to our production team.

Kind regards,

on behalf of

Dr. Alanna Gomes da Silva

Academic Editor

PLOS ONE